# Hippocampome.org 2.0 is a knowledge base enabling data-driven spiking neural network simulations of rodent hippocampal circuits

Diek W Wheeler[1,2]*, Jeffrey D Kopsick[1,3], Nate Sutton[1,2], Carolina Tecuatl[1,2], Alexander O Komendantov[1,2], Kasturi Nadella[1,2], Giorgio A Ascoli[1,2,3]*

[1]Center for Neural Informatics, Structures, & Plasticity, Krasnow Institute for Advanced Study, George Mason University, Fairfax, United States; [2]Bioengineering Department and Center for Neural Informatics, Structures, & Plasticity, College of Engineering and Computing, George Mason University, Fairfax, United States; [3]Interdisciplinary Program in Neuroscience, College of Science, George Mason University, Fairfax, United States

*For correspondence:
dwheele5@gmu.edu (DWW);
ascoli@gmu.edu (GAA)

Competing interest: The authors declare that no competing interests exist.

**Abstract** Hippocampome.org is a mature open-access knowledge base of the rodent hippocampal formation focusing on neuron types and their properties. Previously, Hippocampome.org v1.0 established a foundational classification system identifying 122 hippocampal neuron types based on their axonal and dendritic morphologies, main neurotransmitter, membrane biophysics, and molecular expression (Wheeler et al., 2015). Releases v1.1 through v1.12 furthered the aggregation of literature-mined data, including among others neuron counts, spiking patterns, synaptic physiology, in vivo firing phases, and connection probabilities. Those additional properties increased the online information content of this public resource over 100-fold, enabling numerous independent discoveries by the scientific community. Hippocampome.org v2.0, introduced here, besides incorporating over 50 new neuron types, now recenters its focus on extending the functionality to build real-scale, biologically detailed, data-driven computational simulations. In all cases, the freely down-loadable model parameters are directly linked to the specific peer-reviewed empirical evidence from which they were derived. Possible research applications include quantitative, multiscale analyses of circuit connectivity and spiking neural network simulations of activity dynamics. These advances can help generate precise, experimentally testable hypotheses and shed light on the neural mechanisms underlying associative memory and spatial navigation.

## eLife assessment

The authors have greatly expanded their **important** hippocampome.org resource about rodent hippocampal cell types, their physiological properties, and their interactions. With version 2.0, they make a significant advance in providing a user-friendly means to make computer models of hippocampal circuits. The work is **convincing**, and there are only minor reservations that the figures may be too complex.

## Introduction

Neuroscience knowledge continues to increase every year (*Eke et al., 2022*; *Yeung et al., 2017*), making it challenging for researchers to keep abreast of mounting data and evolving information

even in their own domain of expertise. Large-scale endeavors, such as the Brain Research through Advancing Innovative Neurotechnologies (BRAIN) Initiative (*Insel et al., 2013*) and the European Union's Human Brain Project (*Amunts et al., 2016*), are contributing to this tremendous growth along with the 'long tail' of independent labs and individual scientists (*Ferguson et al., 2014*). A key organizing principle for neuroscience knowledge is the seminal notion of neuron types (*Petilla Interneuron Nomenclature Group et al., 2008*; *Zeng and Sanes, 2017*), which constitute the conceptual 'parts list' of functional circuits. The National Institutes of Health launched the BRAIN Initiative Cell Census Network (BICCN) to help establish a comprehensive reference of the cell type diversity in the human, mouse, and non-human primate brains (*Hawrylycz et al., 2022*). This multi-institution collaboration is already producing innovative results (*Muñoz-Castañeda et al., 2021*) and actionable community resources (*Hawrylycz et al., 2022*).

Hippocampome.org (https://www.hippocampome.org) is an open-access knowledge base of the rodent hippocampal circuit (dentate gyrus, CA3, CA2, CA1, subiculum, and entorhinal cortex) at the mesoscopic level of neuron types (*Wheeler et al., 2015*). This resource has proven popular and effective thanks to the adoption of a simple yet powerful classification system for defining neuron types. Specifically, a key property for the identification of neuron types in Hippocampome.org is the location of axons and dendrites across the subregions and layers of the hippocampal formation. This approach can be broadly extended to classify neurons in other brain regions and neural systems (*Ascoli and Wheeler, 2016*). Focusing on axonal and dendritic distributions provides several considerable advantages. First, these features mediate neuronal connectivity, thus immediately revealing the underlying blueprint of network circuitry (*Rees et al., 2017*). Second, they are widely used in the neuroscience community as a reliable and concrete anchoring signature to correlate electrophysiological and transcriptomic profiles (*DeFelipe et al., 2013*). Third, to coherently classify neuron types, we are not reliant on the inconsistent nomenclature that authors provide (*Hamilton et al., 2017a*). Therefore, starting from the foundational morphology-based identification of 122 neuron types in the first release (version 1.0 or v1.0), Hippocampome.org progressively amassed an increasing amount of complementary data, such as firing patterns (*Komendantov et al., 2019*), molecular expression (*White et al., 2020*), cell counts (*Attili et al., 2022*), synaptic communication (*Moradi et al., 2022*; *Moradi and Ascoli, 2020*), in vivo oscillations (*Sanchez-Aguilera et al., 2021*), and connection probabilities (*Tecuatl et al., 2021b*). In all cases, the public repository provided direct links to the specific peer-reviewed empirical evidence supporting the added knowledge.

Since the inception of Hippocampome.org, we have attempted to maintain the naming styling for already established neuron types by adopting either canonical names, only-cited names, most frequently cited names or hybridizations of cited names, and only as last resort crafting our own names (see Figure 6 in *Hamilton et al., 2017a*). In the entorhinal cortex (EC), where many of the hybridizations occur, we have followed the authors' own definitions for the six layers and the distinction between medial and lateral, when incorprating such terms into Hippocampome.org type names. If a neuron type exists in both medial and lateral entorhinal cortex, then the name is simply prefixed by EC, rather than MEC or LEC. As another example, in the dentate gyrus (DG), we established HIPROM (HIlar Interneuron with PRojections to the Outer Molecular layer) and MOCAP (MOlecular Commissural-Associational Pathway-related axons and dendrites) in the same vein as HIPP (HIlar Perforant Path-associated), MOPP (MOlecular layer Perforant Path-associated), and HICAP (HIlar Commissural-Associational Pathway-related), where the outer two-thirds of the stratum moleculare (SMo) is distinguished by the region intercepted by the Perforant Path from the entorhinal cortex, and the inner one-third (SMi) is characterized by the commissural-associational pathway that often recurrently connects stratum moleculare with the hilus.

Having established a web-based integrated storehouse of hippocampal information, Hippocampome.org also expanded its scope by including data-driven computational models of neuronal excitability and synaptic signaling, as well as ties to community resources such as NeuroMorpho.Org (https://www.neuromorpho.org; *Akram et al., 2018*), SenseLab ModelDB (*McDougal et al., 2017*), CARLsim (*Niedermeier et al., 2022*), and the Allen Brain Atlas (*Jones et al., 2009*). Altogether, these extensions resulted in the emergence of a complete framework in Hippocampome.org v2.0 that makes the original vision of this project, to enable data-driven spiking neural network simulations of rodent hippocampal circuits (*Ascoli, 2010*), finally achievable. The present report thus marks a new phase in the life cycle of this community resource.

One line of research pertaining to the state of simulation readiness of Hippocampome.org involves a real-scale mouse model of CA3 (*Kopsick et al., 2023*) investigating the cellular mechanisms of pattern completion, which includes Pyramidal cells and seven main inhibitory interneuron types. Another avenue of research investigates spatial representation involving in vivo firing grid cells (*Sargolini et al., 2006*), which utilizes Medial Entorhinal Cortex Layer II Stellate cells, two types of pyramidal cells, and three interneuron types. Both lines of research make use of Hippocampome.org parameters for properties such as cell census, Izhikevich models (*Izhikevich, 2003*), synaptic signals, and connection probabilities.

The following 'Description of resource' section begins with a concise, referenced overview of the neural properties collated from Hippocampome.org v1.0 through release v1.12. We then briefly describe the new neuron types and data currently being added in Hippocampome.org v2.0. Next is an abridged summary of the usage and recognition of this online portal in biomedical research. This is followed by an explanation of the latest capabilities of Hippocampome.org v2.0 to search, filter, and download the complete set of computational parameters enabling quantitative connectomic analyses and spiking neural network simulations. The section concludes with an outlook of possible research applications allowed by the expansion of this scientific resource.

## Description of resource
### Characterizing properties of hippocampal neuron types

Hippocampome.org v1.0 (*Wheeler et al., 2015*) established the morphological encoding of axonal and dendritic locations and the main neurotransmitter (glutamate or GABA) as the primary determinants of neuron types in the rodent hippocampal formation. For example, a Dentate Gyrus Basket cell (with name capitalized to indicate a formally defined neuron type) is a GABAergic cell with axon contained in the granular layer and dendrites spanning all dentate gyrus layers (*Figure 1A1-4*). In this framework, two neurons releasing the same neurotransmitter belong to different types if the axon or dendrites of only one of them invades any of the 26 layers across 6 subregions of the hippocampal formation (hippocampome.org/morphology). In other words, neurons of the same type share the same potential inputs, outputs, and excitatory vs. inhibitory function. These properties were initially supplemented with additional empirical evidence for molecular expression of major protein biomarkers (*Figure 1A5*; hippocampome.org/markers) and membrane biophysics (*Figure 1A6-7*; hippocampome.org/electrophysiology).

Many neuronal properties and functionalities were progressively added in 12 subsequent releases (*Table 1*). The numerical sequencing of these Hippocampome.org versions depended on the order of peer-review and publication of the corresponding scientific reports. Here we will describe them instead in logical groupings. The first two updates enhanced the user functionality of the knowledge base. Specifically, v1.1 integrated a web-based interactive thesaurus mapping of synonyms and definitions (*Hamilton et al., 2017a*; hippocampome.org/find-term) to help disambiguate the many terminological inconsistencies in the neuroscience literature (*Shepherd et al., 2019*; *Yuste et al., 2020*). Release v1.2 introduced the capability to browse, search, and analyze the potential connectivity between neuron types (*Rees et al., 2016*; Hippocampome.org/connectivity) as derived from the compiled overlapping locations of all the presynaptic axons and postsynaptic dendrites. Transcriptomic information was greatly expanded in both v1.3 (*Hamilton et al., 2017b*), which incorporated in situ hybridization data from the Allen Brain Atlas (*Lein et al., 2007*), and v1.5 (*White et al., 2020*), which leveraged relational inferences interlinking the region-specific expression of two or more genes.

The quantifications of firing pattern phenotypes, such as rapid adapting spiking, transient stuttering, and persistent slow-wave bursting, in v1.6 (*Komendantov et al., 2019*; hippocampome.org/firing_patterns) were fitted by dynamical systems modeling (*Izhikevich, 2003*) in v1.7 (*Venkadesh et al., 2019*; hippocampome.org/Izhikevich). Although the above properties were largely measured from slice preparations, v1.9 made available measurements from in vivo recordings (*Sanchez-Aguilera et al., 2021*; hippocampome.org/in-vivo). Release v1.10 provided a compendium of cognitive functions linked to specific hippocampal neurons (*Sutton and Ascoli, 2021*; hippocampome.org/cognome), while the v1.11 neuron type census estimated the population counts for each neuron type (*Attili et al., 2022*; hippocampome.org/census). Finally, there are a set of properties pertaining not to individual neuron types but to synaptic connections between a pair of pre- and post-synaptic

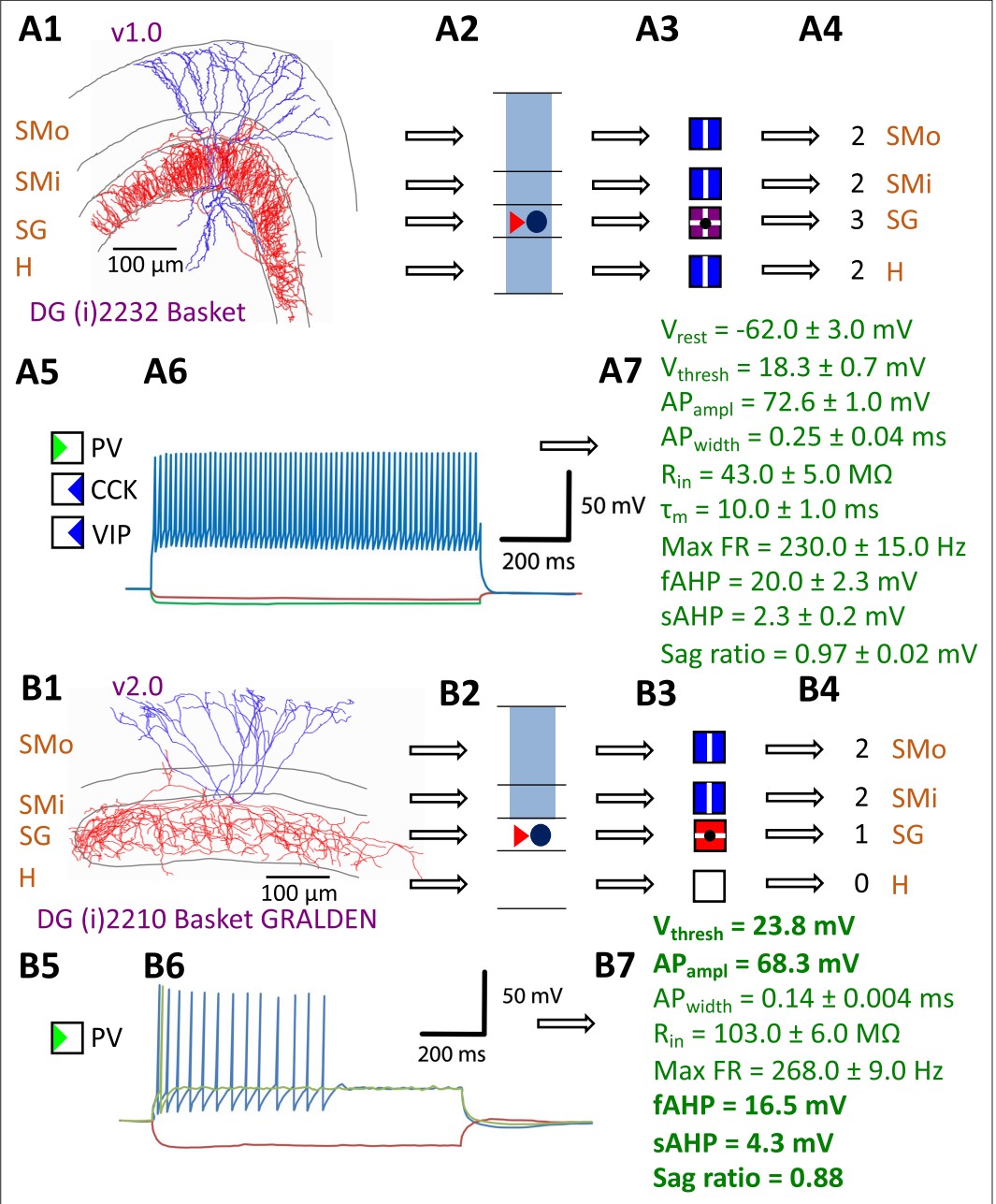

**Figure 1.** Defining neuron types in Hippocampome.org. (**A**) Properties of a Hippocampome.org v1.0 neuron type. (**A1**) Morphology of a Dentate Gyrus (i) 2232 Basket cell (NeuroMorpho.Org cell NMO_34300: Figure S3A in *Hosp et al., 2014*) with axons (red) in *stratum granulosum* (SG) and dendrites (blue) in all four layers. (**A2**) Schematic interpretation of the morphological tracing, where the circle represents the location of the soma in SG, the red triangle the location of the axons in SG, and the blue rectangles the locations of the dendrites in all four layers. (**A3**) Hippocampome.org representation of the morphology, where a blue square with a vertical line (|) indicates dendritic presence in the outer two-thirds of the *stratum moleculare* (SMo) and the inner one-third of the *stratum moleculare* (SMi) and the hilus (H), a purple square with a cross (+) indicates both axonal and dendritic presence in SG, and a black dot (•) indicates the soma location in SG. (**A4**) Hippocampome.org numerical coding of the reconstructed neuron, where 2 indicates the presence of dendrites (in SMo, SMi, and H); and 3 indicates the presence of both axons and dendrites (in SG). (**A5**) Biomarker expressions, where a green triangle indicates positive expression for parvalbumin (PV), and blue triangles indicate negative expression for cholecystokinin (CCK) and vasoactive intestinal polypeptide (VIP). (**A6**) Firing pattern phenotype (non-adapting spiking (NASP); adapted from Figure 1B1 in *Savanthrapadian et al., 2014*). (**A7**) Membrane biophysics values (from Figure 3C and Table 1 in *Lübke et al., 1998*) recorded at 35–37°C. (**B**) Properties for a Hippocampome.org v2.0 neuron

*Figure 1 continued on next page*

*Figure 1 continued*

type. (**B1**) Morphology of a DG (i) 2210 Basket GRALDEN (NeuroMorpho.Org cell NMO_146159: Figure 4S3 in *Vaden et al., 2020*) with red axons in SG and blue dendrites in SMo and Smi. (**B2**) Schematic interpretation of the reconstruction (same symbols as in A2). (**B3-4**) Hippocampome.org representation and numerical coding of the morphology (same symbols as in A3-4). (**B5**) Biomarker expression. (**B6**) Firing pattern phenotype (silence preceded by transient stuttering (TSTUT.SLN); adapted from Figure S4 in *Markwardt et al., 2011*). (**B7**) Membrane biophysics values recorded at room temperature (from Figure 4D in *Vaden et al., 2020*), and at 22 °C (from Figure S4 in *Markwardt et al., 2011*); emboldened values were extracted from the firing pattern trace in B6. Membrane biophysics abbreviations: $V_{rest}$: resting membrane potential; $V_{thresh}$: firing threshold potential; $AP_{ampl}$: action potential amplitude; $AP_{width}$: action potential width; $R_{in}$: input resistance; $\tau_m$: membrane time constant; Max FR: maximum firing rate; fAHP: fast after-hyperpolarizing potential; sAHP: slow after-hyperpolarizing potential; Sag ratio: ratio of the steady-state membrane potential to the minimum membrane potential.

neuron types. In particular, v1.8 calculated the synaptic probabilities and the numbers of contacts per connected pair (*Tecuatl et al., 2021b*; hippocampome.org/syn_probabilities), and v1.4 data mined synaptic physiology (*Moradi and Ascoli, 2020*; hippocampome.org/synaptome), with conductance, time constant, and short-term plasticity values normalized by age, temperature, species, sex, and

**Table 1.** Added knowledge and functioning in Hippocampome.org releases v1.1–12.

| Version | Contribution | Article |
|---|---|---|
| v1.1 | • definitions for terms and phrases relevant to Hippocampome.org | *Hamilton et al., 2017a* |
| v1.2 | • clickable connectivity matrix<br>• interactive connectivity navigator Java applet<br>• searching by connectivity | *Rees et al., 2016* |
| v1.3 | • downloadable list of ABA predictions of marker expressions<br>• utility for viewing the effects of thresholds on ABA marker expression predictions | *Hamilton et al., 2017b* |
| v1.4 | • access to the synapse knowledge base | *Moradi and Ascoli, 2020* |
| v1.5 | • relational biomarker expression inferences | *White et al., 2020* |
| v1.6 | • firing pattern phenotypes<br>• clickable firing pattern matrix<br>• clickable firing pattern parameters matrix<br>• search by firing pattern<br>• search by firing pattern parameter | *Komendantov et al., 2019* |
| v1.7 | • Izhikevich models<br>• clickable Izhikevich model parameters matrix<br>• downloadable single-neuron parameter files<br>• downloadable single-neuron CARLSim4 simulation files<br>• ability to perform single-neuron simulations of the firing patterns | *Venkadesh et al., 2019* |
| v1.8 | • clickable/downloadable neurite lengths matrix<br>• clickable/downloadable somatic path distances matrix<br>• clickable/downloadable numbers of potential synapses matrix<br>• clickable/downloadable numbers of contacts matrix<br>• clickable/downloadable connection probabilities matrix | *Tecuatl et al., 2021b* |
| v1.9 | • clickable matrix for in vivo recordings | *Sanchez-Aguilera et al., 2021* |
| v1.10 | • Cognome knowledge base of spiking neural circuit functions and network simulations of the hippocampal formation | *Sutton and Ascoli, 2021* |
| v1.11 | • clickable matrix of neuron type census values for rat and mouse | *Attili et al., 2022* |
| v1.12 | • clickable/downloadable matrices of synaptic physiology parameter values ($g$, $\tau_d$, $\tau_r$, $\tau_f$, $U$) for combinations of species, sex, age, temperature, and recording mode | *Moradi et al., 2022* |

recording method in v1.12 (*Moradi et al., 2022*; hippocampome.org/synapse), leveraging machine learning and a phenomenological model (*Tsodyks et al., 1998*).

## Expanding the catalog of neuron types and properties from Hippocampome.org v1.x to v2.0

The Hippocampome.org framework to classify neuron types and collate their properties allows agile content updates as new data are continuously reported in the peer-reviewed literature. For example, the description of a parvalbumin-positive dentate gyrus GABAergic interneuron with axon contained in the granular layer and dendrites invading the molecular layer but not the hilus (*Vaden et al., 2020*) supported the definition of a new neuron type (*Figure 1B1-5*), referred to in Hippocampome.org v2.0 as DG Basket GRALDEN (GRAnular Layer DENdrites) cell. Moreover, such an identification made it possible to unequivocally ascribe to this neuron type previously

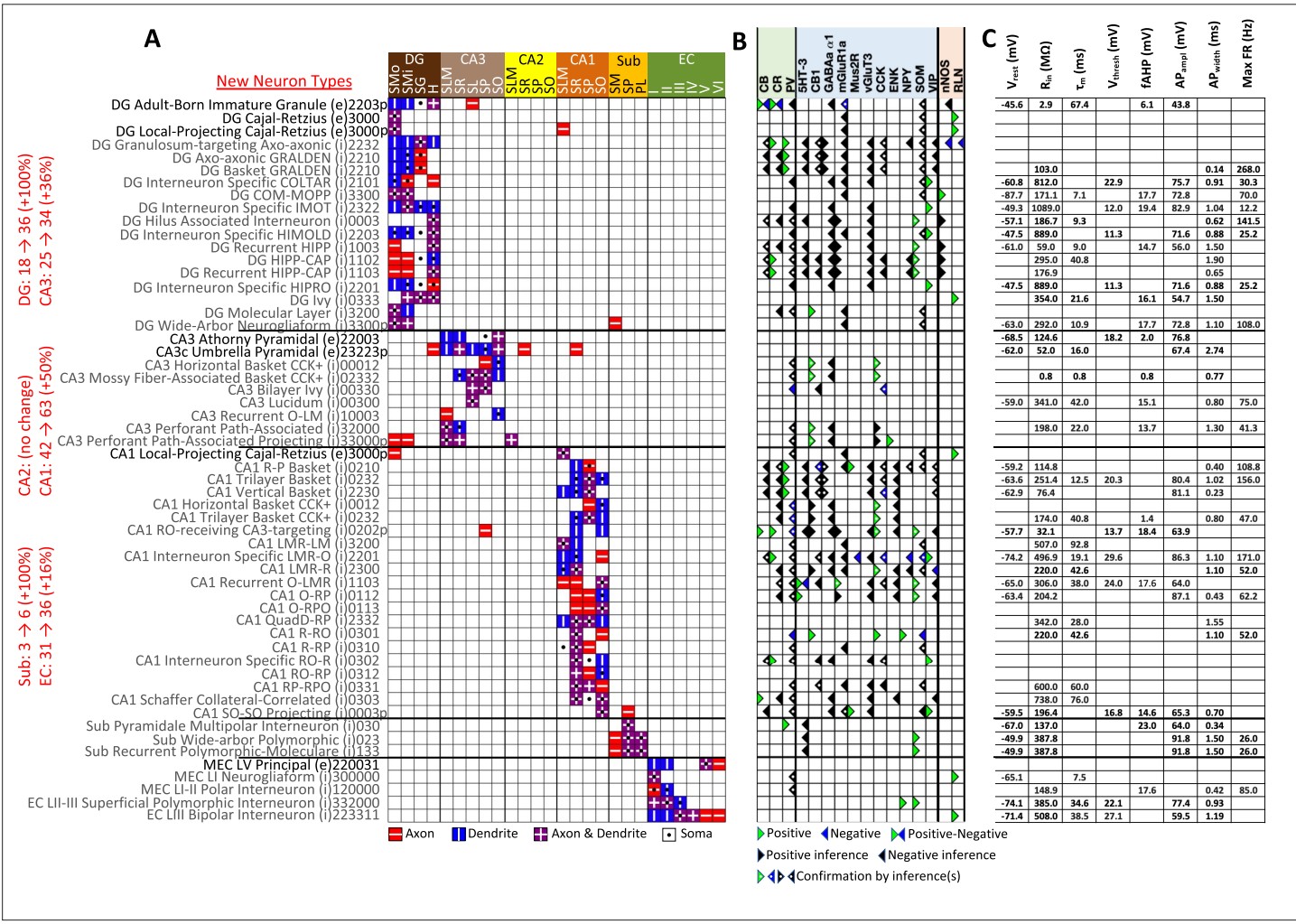

**Figure 2.** New neuron types added to Hippocampome.org v2.0. (**A**) Morphology encodings of the 56 new neuron types that extend the 124 types in Hippocampome.org v1.12. (Left) Increase in number of neuron types for each subregion. For the neuron type names, excitatory types (e) are in black font and inhibitory types (i) are in gray font. The 3–6 digit numbers encode the patterns of axons and dendrites in the layers of the home subregion of the neuron type: 0=no axons or dendrites, 1=only axons, 2=only dendrites, 3=both axons and dendrites. A "p" indicates that the neuron types projects across into other subregions. (**B**) Biomarker expressions of the neuron types. (**C**) Membrane biophysics values for the neuron types. Morphology abbreviations: DG: dentate gyrus; SMo: outer two-thirds of *stratum moleculare*; SMi: inner one-third of *stratum moleculare*; SG: *stratum granulosum*; H: hilus; SLM: *stratum lacunosum-moleculare*; SR: *stratum radiatum*; SL: *stratum lucidum*; SP: *stratum pyramidale*; SO: *stratum oriens*; Sub: subiculum; PL: polymorphic layer; EC: entorhinal cortex. Marker abbreviations: CB: calbindin; CR: calretinin; PV: parvalbumin; 5HT-3: serotonin receptor 3; CB1: cannabinoid receptor type 1; GABAa α1: GABA-a alpha 1 subunit; mGluR1a: metabotropic glutamate receptor 1 alpha; Mus2R: muscarinic type 2 receptor; vGluT3: vesicular glutamate transporter 3; CCK: cholecystokinin; ENK: enkephalin; NPY: neuropeptide Y; SOM: somatostatin; VIP: vasoactive intestinal polypeptide; nNOS: neuronal nitric oxide synthase; RLN: reelin. Membrane biophysics abbreviations: see *Figure 1*.

reported electrophysiological characteristics (*Figure 1B6-7*; *Markwardt et al., 2011*). Comprehensive literature mining following the same process expanded the Hippocampome.org v2.0 catalog with 56 new neuron types across 5 of the 6 subregions of the hippocampal formation (*Figure 2*), including axonal-dendritic morphological patterns (*Figure 2A*), molecular expression (*Figure 2B*), and membrane biophysics (*Figure 2C*).

Besides identifying new neuron types, the Hippocampome.org classification system also allows the ongoing accumulation of new properties onto existing neuron types as well as the reconciliation of fragmented descriptions from scientific publications (*Figure 3*). For instance, converging evidence indicates that Entorhinal Cortex Layer III Pyramidal cells have axonal projections in all layers of CA1 (*Deller et al., 1996*; *Takács et al., 2012*), not just in stratum lacunosum-moleculare (SLM) as originally reported (*Steward, 1976*). Hippocampome.org v2.0 captures both the new extracted knowledge and the corresponding experimental evidence (*Figure 3A*). The annotation of neuron type-specific firing phases relative to in vivo oscillations in v1.9 highlighted a clear distinction between Superficial and Deep CA1 Pyramidal cells (*Sanchez-Aguilera et al., 2021*). The present release enriches that description with accompanying novel molecular markers (*Figure 3B1*), membrane biophysics values (*Figure 3B2*), and differential connectivity with other subregions and neuron types (*Figure 3B3*). Similarly, numerous additional firing patterns (*Figure 3C*) have been datamined for existing neuron types, such as adapting spiking in CA1 Oriens-Bistratified cells (*Craig and McBain, 2015*), non-adapting spiking in CA3 Basket Cholecystokinin-positive (CCK+) cells (*Szabadics and Soltesz, 2009*) or transient stuttering in CA1 Radiatum Giant cells (*Kirson and Yaari, 2000*). Notably, this includes a novel phenotype, transient stuttering followed by persistent stuttering (TSTUT.PSTUT) in CA1 Interneuron Specific O-targeting QuadD cells (*Chamberland et al., 2010*). With this report, we also release new differential connection probabilities to various CA1 neuron type targets from traditional CA3 Pyramidal cells vs. CA3c Pyramidal cells (*Figure 3D*) and from Dentate Gyrus Granule cells to mossy fiber CA3 targets (*Table 2*).

## Quantifying the content and impact of Hippocampome.org

Over the course of subsequent releases, we have measured Hippocampome.org content using two metrics. The number of pieces of knowledge (PoK) tallies the distinct units of structured information, such as the statements that Dentate Gyrus Granule cell axons invade the hilus or that CA1 Basket cells express parvalbumin. The pieces of evidence (PoE) are specific excerpts of peer reviewed publications (portion of text, figure, or table) or database entries (e.g. from the Allen Brain Atlas) always linked to each PoK. Both PoK and PoE continued to grow with successive releases of Hippocampome.org (*Figure 4A*). Notably, the largest increases in PoK and PoE were related to synaptic properties (*Moradi and Ascoli, 2020*; *Tecuatl et al., 2021b*; *Moradi et al., 2022*). Specifically, the data underlying synaptic physiology and connection probabilities were supported by over 23,000 PoE and yielded a remarkable 500,000 PoK thanks to the normalized collection of signaling and short-term plasticity modeling parameters for multiple combinations of experimental conditions.

To assess community usage of Hippocampome.org, we tracked the number of citations of the original publication (*Wheeler et al., 2015*) and of the subsequent versions (*Figure 4B*), separating simple references from actual employment of information extracted from Hippocampome.org for secondary analyses (*Table 3*). At the time of this writing, year 2021 proved to be the most prolific citation-wise; however, more than a third of the releases (v1.8–12) appeared after 2021 and most PoK were added in 2022, so usage could potentially accelerate further in coming years. An early application of Hippocampome.org-sourced data used subthreshold biophysical measures, such as input resistance and membrane time constant, for multicompartmental models of signal integration and extracellular field generation (*Gulyás et al., 2016*). That study concluded that somatic and proximal dendritic intracellular recordings in pyramidal cells and calretinin-positive interneurons, in particular, do not capture a sizable portion of the synaptic inputs. As a recent usage example, another lab employed Hippocampome.org as the primary information resource for neuron types in DG, CA3, and CA1 (*Schumm et al., 2022*). They discovered that mild traumatic brain injury, in the form of alterations in spike-timing-dependent plasticity, may affect the broadband power in CA3 and CA1 and the phase coherence between CA3 and CA1.

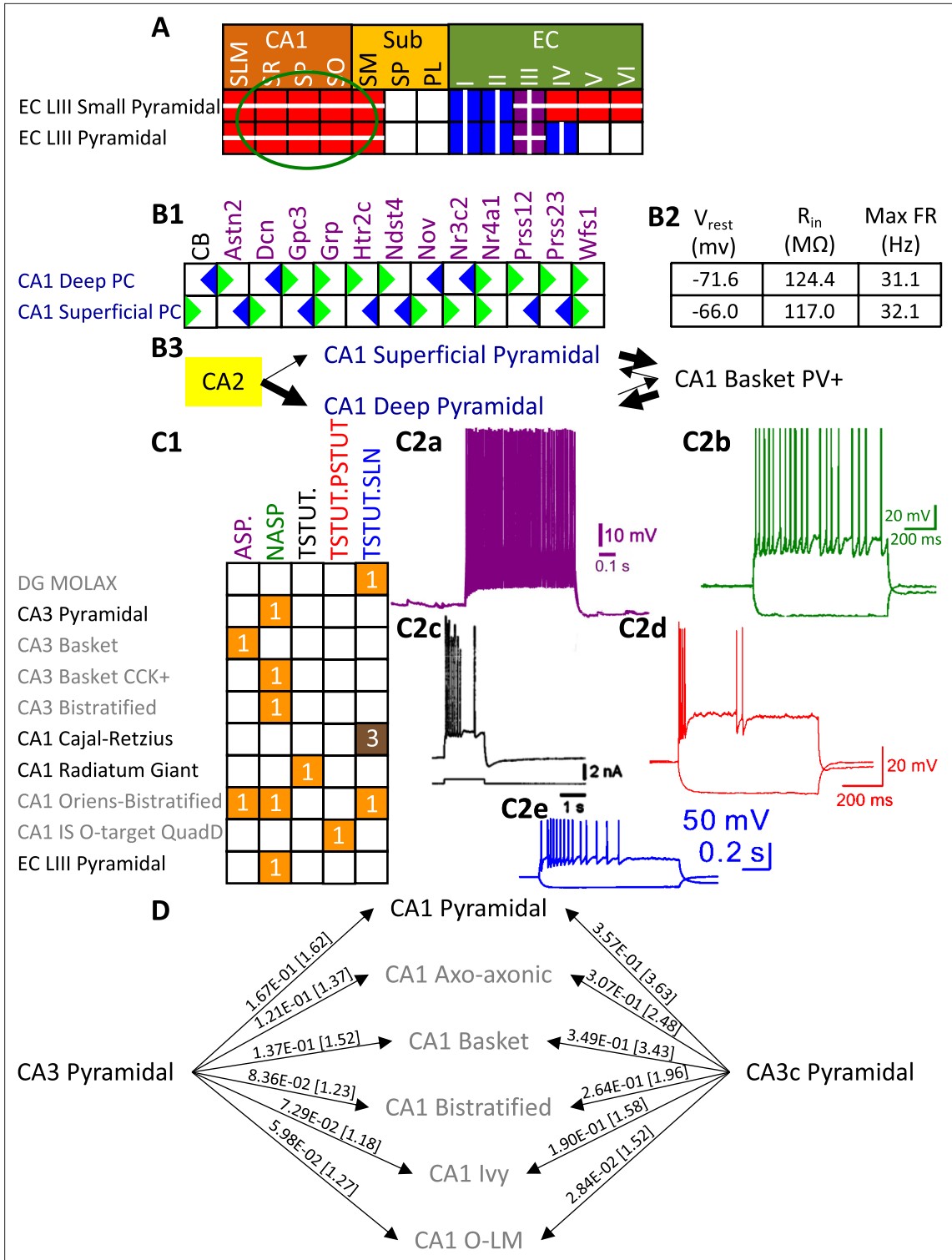

**Figure 3.** Extensions to the neuronal properties of Hippocampome.org v1.x. (**A**) Additions to the axonal projections (circled in green) for two v1.0 neuron types, based on information derived from Figure 2b in *Deller et al., 1996*. (**B1**) Biomarker expressions for the two CA1 Pyramidal sub-types added in Hippocampome.org v1.9 (*Sanchez-Aguilera et al., 2021*). (**B2**) Membrane biophysics values for the two sub-types. (**B3**) CA2 projects preferentially to the deep sublayer of CA1 (*Kohara et al., 2014*). More perisomatic parvalbumin-positive (PV+) GABAergic boutons are found at CA1 Deep Pyramidal cells (*Valero et al., 2015*). CA1 Superficial Pyramidal cells form more frequent connections to PV + CA1 Basket cells, and PV + CA1 Basket cells form significantly more perisomatic axon terminals on CA1 Deep Pyramidal cells (*Lee et al., 2014*). (**C1**) Additions to the firing pattern phenotypes of v1.0 neuron types. (**C2a**) Example of adapting spiking (ASP.) in a CA1 Oriens-Bistratified cell (adapted from Figure 4B in *Craig and*

*Figure 3 continued on next page*

*Figure 3 continued*

**McBain, 2015**). (**C2b**) Example of non-adapting spiking (NASP) in a CA3 Basket CCK + cell (adapted from Figure 3A in **Szabadics and Soltesz, 2009**). (**C2c**) Example of transient stuttering (TSTUT.) in a CA1 Radiatum Giant cell (reproduced from Figure 2Bb in **Kirson and Yaari, 2000**). (**C2d**) Example of transient stuttering followed by persistent stuttering (TSTUT.PSTUT) in a CA1 Interneuron Specific O-targeting QuadD (adapted from Figure 2D in **Chamberland et al., 2010**). (**C2e**) Example of silence preceded by transient stuttering (TSTUT.SLN) in a DG MOLAX cell (adapted from Figure S2c in **Lee et al., 2016**). (**D**) Synaptic probabilities for projecting connections and the corresponding number of contacts in brackets between the two CA3 Pyramidal neuron types and a selection of CA1 neuron types. Morphology abbreviations: see **Figure 2**. Marker abbreviations: CB: calbindin; Astn2: astrotactin 2; Dcn: decorin; Gpc3: glypican 3; Grp: gastrin releasing peptide; Htr2c: 5-hydroxytryptamine receptor 2 c; Ndst4: N-deacetylase and N-sulfotransferase 4; Nov: nephroblastoma overexpressed; Nr3c2: nuclear receptor subfamily 3 group C member 2; Nr4a1: nuclear receptor subfamily 4 group A member 1; Prss12: serine protease 12; Prss23: serine protease 23; Wfs1: wolframin ER transmembrane glycoprotein. Membrane biophysics abbreviations: see **Figure 1**.

## From experimental data to biologically realistic computational models

Several key neural properties collated into Hippocampome.org have gradually transformed the site from an organized repository of hippocampal knowledge to a computational framework for launching real-scale neural network simulations. Specifically, building a data-driven circuit model of a neural system (such as the hippocampal formation or portion thereof) requires four essential quantities besides the full list of neuron types (**Bahmer et al., 2023**; **DePasquale et al., 2023**): (i) the number of neurons in each type; (ii) the input-output response function for each neuron type; (iii) the connection probability for each pair of interacting neuron types; and (iv) the unitary synaptic signals for each pair of connected neuron types (**Figure 5**). Of those quantities, (i) and (ii) are neuron type properties, while (iii) and (iv) are properties of directional connections, defined as a distinct pair of a presynaptic and a postsynaptic neuron type. Moreover, (i) and (iii) are structural features, while (ii) and (iv) are electro-physiological ones.

Hippocampome.org v1.11 provides estimates of the number of neurons in each neuron type (i) for both rats and mice (**Figure 5A**). These values were derived in a two-step process (**Attili et al., 2020**): first, literature mining extracted suitable quantitative relations such as the cellular density in a given layer (**Attili et al., 2019**), the total count of neurons expressing a certain gene, or the fraction of sampled cells with a particular morphology; second, numerical optimization of the corresponding equations yielded a complete census for all neuron types. As of v1.7, Hippocampome.org represents

**Table 2.** Probabilities of connection and number of contacts per connected pair from DG Granule cell to mossy fiber targets in CA3.

| Postsynaptic neuron type | Probability | # contacts |
|---|---|---|
| CA3 Pyramidal | 1.11E-04 | 1.08 |
| CA3c Pyramidal | 3.91E-04 | 1.31 |
| CA3 Spiny Lucidum Dentate-Projecting | 5.89E-04 | 1.69 |
| CA3 Mossy Fiber-Associated ORDEN | 4.44E-04 | 1.27 |
| CA3 Basket | 6.55E-04 | 1.50 |
| CA3 Basket CCK+ | 2.14E-04 | 1.16 |
| CA3 Ivy | 3.35E-04 | 1.29 |
| CA3 Mossy Fiber-Associated | 3.78E-05 | 1.04 |
| CA3 LMR-Targeting | 1.31E-04 | 1.21 |
| CA3 Lucidum ORAX | 2.62E-04 | 1.19 |
| CA3 Lucidum-Radiatum | 3.25E-04 | 1.13 |
| CA3 Axo-Axonic | 7.56E-04 | 1.50 |
| CA3 Bistratified | 8.25E-04 | 1.45 |
| CA3 QuadD-LM | 2.91E-04 | 1.25 |

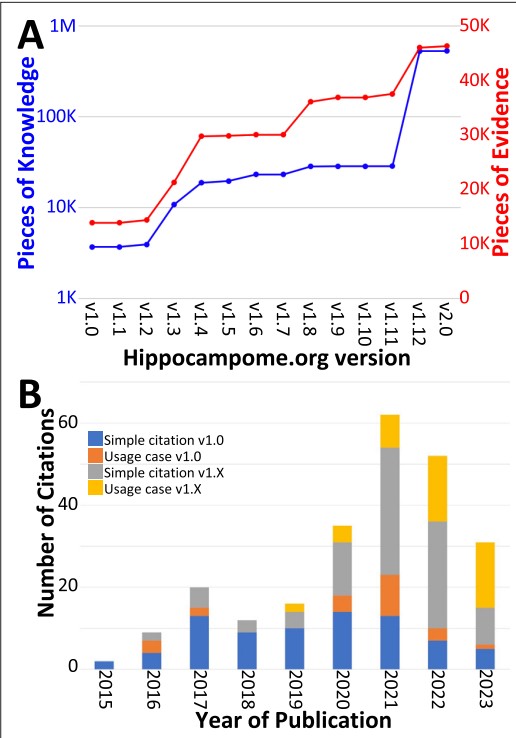

**Figure 4.** Trends in Hippocampome.org data, knowledge, citations, and usage since v1.0. (**A**) Increase in pieces of knowledge (blue) and evidence (red) with Hippocampome.org version number. (**B**) Number of citations, in which the publication is simply referenced (blue and gray portions), and usage cases, in which the citing work makes use of the information contained within the Hippocampome.org-related work (orange and yellow portions), by year.

the neuronal input-output response function (ii) in the form of single- and multi-compartment Izhikevich models (*Figure 5B*) fitted by evolutionary algorithms to accurately reproduce the observed firing behavior of each neuron type (*Venkadesh et al., 2018*). For v1.8, the connection probability (iii) from one neuron type to another (*Figure 5C*) was computed from measurements of the appropriate axonal and dendritic lengths in each invaded subregion and layer (hippocampome.org/A-D_lengths). Additionally, users can also access the presynaptic and postsynaptic path distances from the respective somata ( hippocampome.org/soma_distances) and the number of contacts per connected neuron pairs (hippocampome.org/num_contacts). As for the synaptic communication between neurons (iv), Hippocampome.org v1.12 adopted the Tsodyks-Pawelzik-Markram formulation, representing unitary signals and short-term plasticity with five constants for each directional pair of interacting neuron types: the synaptic conductance, decay time, recovery time, facilitation time, and the utilization ratio (*Tsodyks et al., 1998*; *Moradi et al., 2022*). Once again, these parameters were fitted from the experimental data (*Cutsuridis et al., 2018*) employing deep learning to account for (and predict the effects of) numerous experimental variables (*Figure 5D*), including species (rat vs. mouse), sex (male vs. female), age (young vs. adult), recording temperature (room vs. body), and clamping configuration (voltage vs. current).

The above description underscores the crucial interconnectedness of individually measured neuronal properties forming a cohesive whole in Hippocampome.org (*Figure 6*). In particular, normalized simulation parameters (e.g. the sensitivity of recovery variable in Izhikevich models) are derived from quantitative experimental measurements, such as the spiking adaptation rate (*Figure 6a*). Those in turn are linked to an identified neuron type based on qualitative features, like calbindin expression or laminar distribution of axons and dendrites. In addition to enabling computational applications as described below, such integration also allows the meta-analysis of correlations between morphological features, molecular profiles, electrophysiological properties, and dynamic circuit functions. At the same time, several components of Hippocampome.org are also synergistically linked to external community resources (*Figure 6B*). For example, each neuron page links out to all three-dimensional morphological reconstructions of the same cell type available in NeuroMorpho.Org (*Ascoli et al., 2007*), and selected data from NeuroMorpho.Org were used to compute axonal and dendritic length and connection probabilities. Each neuron page also links out to all computational models (including Hodgkin-Huxley, stochastic diffusion, mean firing rate, etc.) involving the same cell type on ModelDB (*McDougal et al., 2017*), while conversely ModelDB includes the Izhikevich models for all Hippocampome.org neuron types. Moreover, simulation parameters from Hippocampome.org are exportable to the CARLsim simulation environment (*Nageswaran et al., 2009*), enabling fast execution of spiking neural network models optimized for GPUs. Furthermore, Hippocampome.org harnessed data from the Allen Brain Atlas (*Lein et al., 2007*) to infer gene expression for principal neurons and cell densities for use in the neuron type census.

**Table 3.** Examples of independent studies utilizing unique neuronal properties from Hippocampome.org v1.0.

| Article | Usage |
|---|---|
| *Gulyás et al., 2016* | Lists of subthreshold physiological properties for multicompartmental modeling |
| *Skene and Grant, 2016* | Catalog of CA1 Interneuron types |
| *Faghihi and Moustafa, 2017* | Diversity of hippocampal neuron types and morphological neuronal features |
| *Puighermanal et al., 2017* | Biomarker expression in CA1 interneurons |
| *Depannemaecker et al., 2020* | Parameter values for a model of synaptic neurotransmission |
| *Ecker et al., 2020* | Evidence that CA1 interneurons express multiple overlapping chemical markers |
| *Hunsberger and Mynlieff, 2020* | Cell identification based on firing properties |
| *Schumm et al., 2020* | Directionality of connections in the hippocampus |
| *Aery Jones et al., 2021* | Local connectivity of CA1 PV + interneurons |
| *Ciarpella et al., 2021* | Lists of hippocampal genes |
| *Luo et al., 2021* | Confirmation of multiple hippocampal neuron types |
| *Mehta et al., 2021* | Connectome model inspired by entorhinal-CA1 circuit |
| *Obafemi et al., 2021* | Principal channels of information processing are DG Granule cells and CA1-3 Pyramidal cells |
| *Sáray et al., 2021* | Membrane biophysics values for CA1 Pyramidal cells |
| *Smith et al., 2021* | Omni-directionality of axons of CA1 Pyramidal cells |
| *Venkadesh and Van Horn, 2021* | Example of a brain region's mesoscopic structural connectivity |
| *Walker et al., 2021* | Reference to morphological and molecular characteristics of hippocampal principal cells and interneurons |
| *Wynne et al., 2021* | Example brain region with a variety of cell types |
| *Kopsick et al., 2023* | Utilize accumulated knowledge as the basis for simulations |
| *Schumm et al., 2022* | Hippocampal morphology, biomarker expression, connectivity, and typing of neurons |
| *Zagrean et al., 2022* | Diversity of hippocampal neuronal types and their properties |

To facilitate construction of spiking neural network simulations, Hippocampome.org v2.0 also includes a new graphical user interface (GUI). With this GUI, users can download sets of simulation parameter values for arbitrarily selected neuron types, a subregion of interest, or the whole hippocampal formation (*Figure 7*). The sets consist of files for the instantiation of CARLsim simulations and a comma-separated values (CSV) spreadsheet of parameters for use in a different simulation environment of the user's choice. For the convenience of users interested in simplified circuit models, Hippocampome.org informally ranks the importance of each neuron type in terms of the functional role it plays in the hippocampal circuitry from 1 (essential) to 5 (dispensable). For instance, a user may choose to simulate only the canonical, or rank 1, neuron types of the tri-synaptic circuit and entorhinal cortex, consisting of Dentate Gyrus Granule, CA3 Pyramidal, CA1 Pyramidal, and Medial Entorhinal Cortex Layer II Stellate cells. When Hippocampome.org is missing a parameter value due to insufficient experimental evidence, the GUI exports a default value clearly indicating so in the downloadable files. For missing Izhikevich and synaptic signaling parameters, the default values are those provided by the CARLsim simulation environment. For missing synaptic probabilities, Hippocampome.org

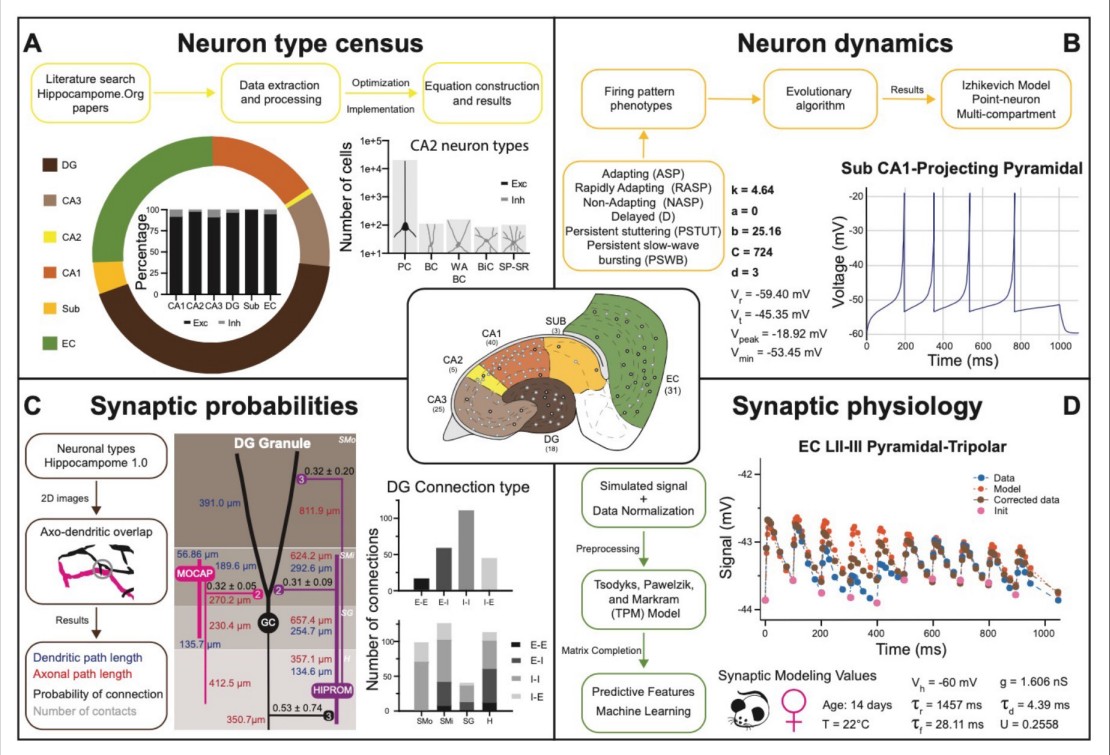

**Figure 5.** Transitional knowledge enabling Hippocampome.org to support spiking neural network simulations. Center: General diagram of the hippocampal formation and the number of cell types in Hippocampome.org v1.0. (**A**) Neuron type census. Top: General pipeline for obtaining cell counts for specific collections of neurons from the peer reviewed literature. Left: Neuron count proportions for the different subregions of the hippocampal formation. Insert: Normalized neuron counts for the inhibitory vs. excitatory balance by subregion. Right: Neuron counts for five identified CA2 neuron types (green schematic: excitatory, red schematic: inhibitory). (**B**) Neuron dynamics. Top: General pipeline for obtaining Izhikevich models to reproduce the firing pattern phenotypes from peer reviewed data. Right: Simulated firing pattern from a Sub CA1-Projecting Pyramidal cell in response to a 250 pA current injection pulse lasting 1 s. Izhikevich model parameters are shown in bold and the membrane biophysics properties are shown in regular font. (**C**) Synaptic probabilities. Left: General pipeline for obtaining the connection probabilities, number of contacts, and dendritic and axonal path lengths from 2D reconstructions. Middle: Example of a connectivity diagram of a DG Granule cell and two interneurons across the different parcels of DG. Probabilities of connection (mean ± SD) are shown in black, numbers of contacts in gray, dendritic path lengths in blue, and axonal lengths in red. Right top: Total number of connections within DG by connection type. Right bottom: Breakdown of the total number of connections by parcel and connection type. (**D**) Synaptic physiology. Left: General pipeline for obtaining normalized synaptic parameters from paired recordings with a TPM model. Right top: Digitized synaptic data between two EC LII-III Pyramidal-Tripolar cells. Experimental data are shown in blue, initiation synaptic points in pink, model data in orange, and corrected data in green. Right bottom: Simulated modeling conditions, electrophysiological parameters, and TPM parameters. Abbreviations by panel: (**A**) PC: Pyramidal cell; BC: Basket cell; WA BC: Wide-Arbor Basket cell; BiC: Bistratified cell; Exc: excitatory; Inh: inhibitory. (**B**) $V_r$: resting membrane potential; $V_t$: firing threshold potential; $V_{peak}$: spike cutoff potential; $V_{min}$: post-spike reset potential. (**C**) E-E: excitatory-excitatory; E-I: excitatory-inhibitory; I-I: inhibitory-inhibitory; I-E: inhibitory-excitatory. (**D**) $V_h$: holding potential; $\tau_r$: recovery time constant; $\tau_f$: facilitation time constant; g: conductance; $\tau_d$: deactivation time constant; U: utilization ratio.

precomputes values averaged by connection type, namely excitatory-excitatory (0.0117), excitatory-inhibitory (0.0237), inhibitory-excitatory (0.00684), and inhibitory-inhibitory (0.00423).

## Potential applications to connectomic analyses and spiking neural networks simulations

Hippocampome.org v2.0 enables the multiscale analysis of circuit connectivity (*Figure 8*). At the highest echelon are the connections between hippocampal subregions, which are comprised of the mesoscopic level potential connectivity between individual neuron types (*Figure 8A*). Expanding, for example, upon the 147 connections between the dentate gyrus and CA3 neuron types reveals all the connections every individual neuron type forms with the other neuron types within and across the subregions (*Figure 8B*). Zooming in onto a single neuron type each from the dentate gyrus and CA3, it is possible to quantify the efferent and afferent connections with other neuron types from throughout the hippocampal formation in terms of synaptic probabilities and number of neuronal

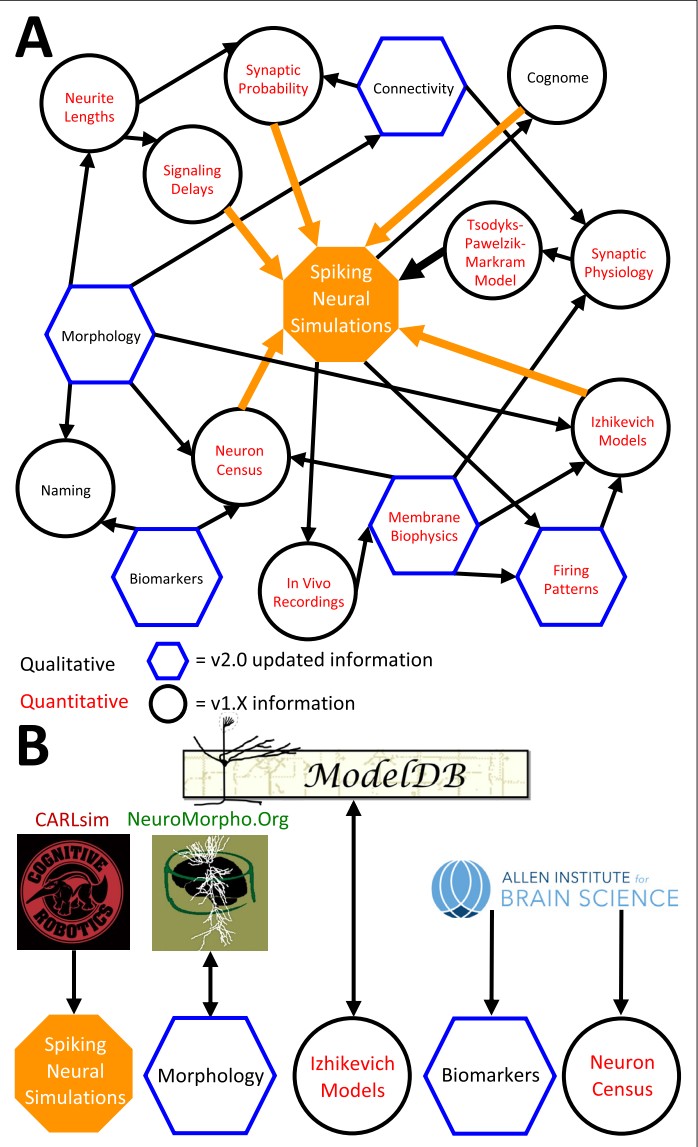

**Figure 6.** Hippocampome.org data provenance. (**A**) The internal web of constituent neuron-type properties (black thin arrows) that ultimately contribute to the instantiation of spiking neural simulations (orange thick arrows). Properties described qualitatively, such as morphological presence of axons in a layer or molecular biomarker expressions, are in black font. Properties described by quantitative values, such as membrane biophysics and neurite lengths, are in red font. Properties with v2.0 updated information, such as connectivity and firing pattern phenotypes, are depicted by blue hexagons, and v1.x information, such as Izhikevich modeling parameter values and neuron-type census values, is visualized by black circles. (**B**) External resources that contribute data to and receive data from Hippocampome.org (the ModelDB logo has been modified from the original).

partners (*Figure 8C*). Diving even deeper into the isolated connection between two neuron types, such as the mossy fiber contacts from Dentate Gyrus Granule cells to CA3 Basket cells, expands the connectivity analysis to several physiological factors affecting neuronal communication: the subcellular location of the synaptic contact (e.g. soma in *stratum pyramidale* and proximal dendrites in *stratum lucidum*), the transfer function (product of synaptic conductance and decay time constant), the in vivo firing rate of the presynaptic neuron type, and the relationship between input current and resulting output spiking frequency (F-I curve) of the post-synaptic neuron (*Figure 8D*).

Release of v2.0 makes the original objective of Hippocampome.org, to enable data-driven spiking neural network simulations of rodent hippocampal circuits (*Ascoli, 2010*), finally achievable. An ongoing line of research in this regard focuses on a real-scale mouse model of CA3, with the eventual

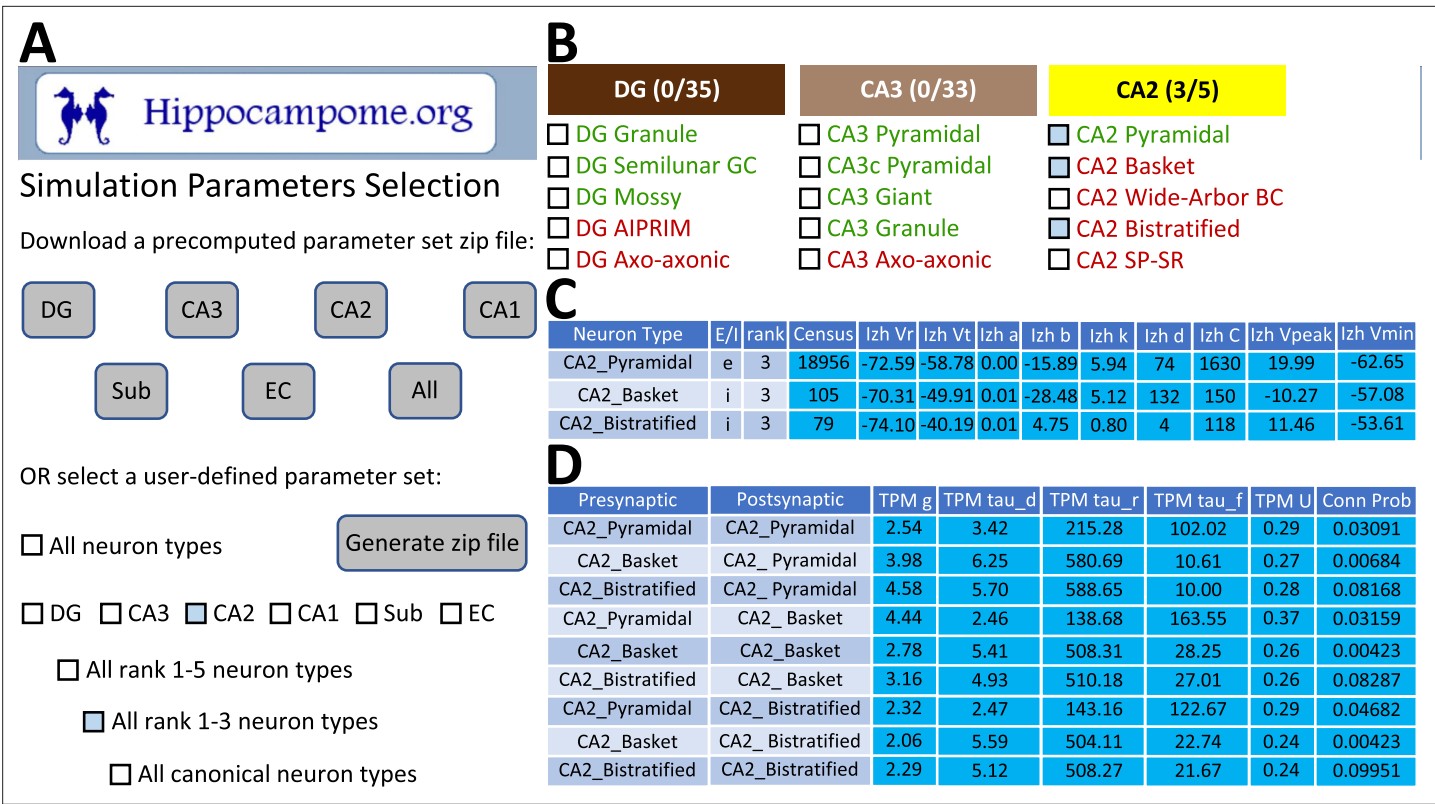

**Figure 7.** CARLsim simulation parameters selection and file generation interface. (**A**) The user chooses which subset of the available neuron types to include in the generated downloadable parameter file. Neuron types can be selected (check boxes and gray highlights) either individually or by groupings, such as by subregion and/or by importance rank. (**B**) Representative user selection. (**C**) Downloadable neuron-level parameters. (**D**) Downloadable connection-level parameters.

goal of investigating the cellular mechanisms of pattern completion. Initial work included excitatory Pyramidal cells and seven main inhibitory interneuron types: Axo-axonic cells (AAC), Basket cells (BC), Basket Cholecystokinin-positive cells (BC CCK+), Bistratified cells (BiC), Ivy cells, Mossy Fiber-Associated ORDEN (MFA ORDEN) cells, and QuadD-LM (QuadD) cells (*Kopsick et al., 2023*). Use of Hippocampome.org parameters for cell census, Izhikevich models, synaptic signals, and connection probabilities resulted in robust, realistic, rhythmic resting state activity for all neuron types (*Figure 9A*). Building off the constructed network of CA3, we seek to understand how the neuron type circuit may allow for the formation of cell assemblies that correspond to distinct memories. Additionally, we will evaluate the network's pattern completion capabilities when presented with degraded input patterns (*Guzman et al., 2021*). Furthermore, associations between memories in CA3 may be encoded by cells shared between cell assemblies (*Gastaldi et al., 2021*). Therefore, we will investigate how cell assembly size and overlap may impact memory storage and recall and the role of different neuron types in associating cell assemblies with one another.

Another pursuit using a spiking neural network seeks to replicate the spatial representation in grid cells (*Sargolini et al., 2006*), modeled utilizing Hippocampome.org Medial Entorhinal Cortex Layer II Stellate cells (SC), and supported by various GABAergic interneuron types (*Dhillon and Jones, 2000*): Axo-axonic (AA), Basket cells (BC), and Entorhinal Cortex Layer II Basket-Multipolar cells (BC MP). This study aims to reproduce the in vivo firing of these neuron types as a virtual rodent explores an open field (*Figure 9B*). Different theories offer potential neural mechanisms underlying the grid cell phenomenon, but it remains challenging to test them comprehensively for anatomical and electrophysiological consistency with experimental data (*Sutton and Ascoli, 2021*; *Zilli, 2012*). Our work, which is in preparation for publication, demonstrates that a spiking neural network implementation of one such theory, the continuous attractor model, generates grid field activity highly compatible with that measured in behaving animals when utilizing Hippocampome.org model parameters. While preliminary, these illustrative examples highlight the potential of Hippocampome.org enabled data-driven

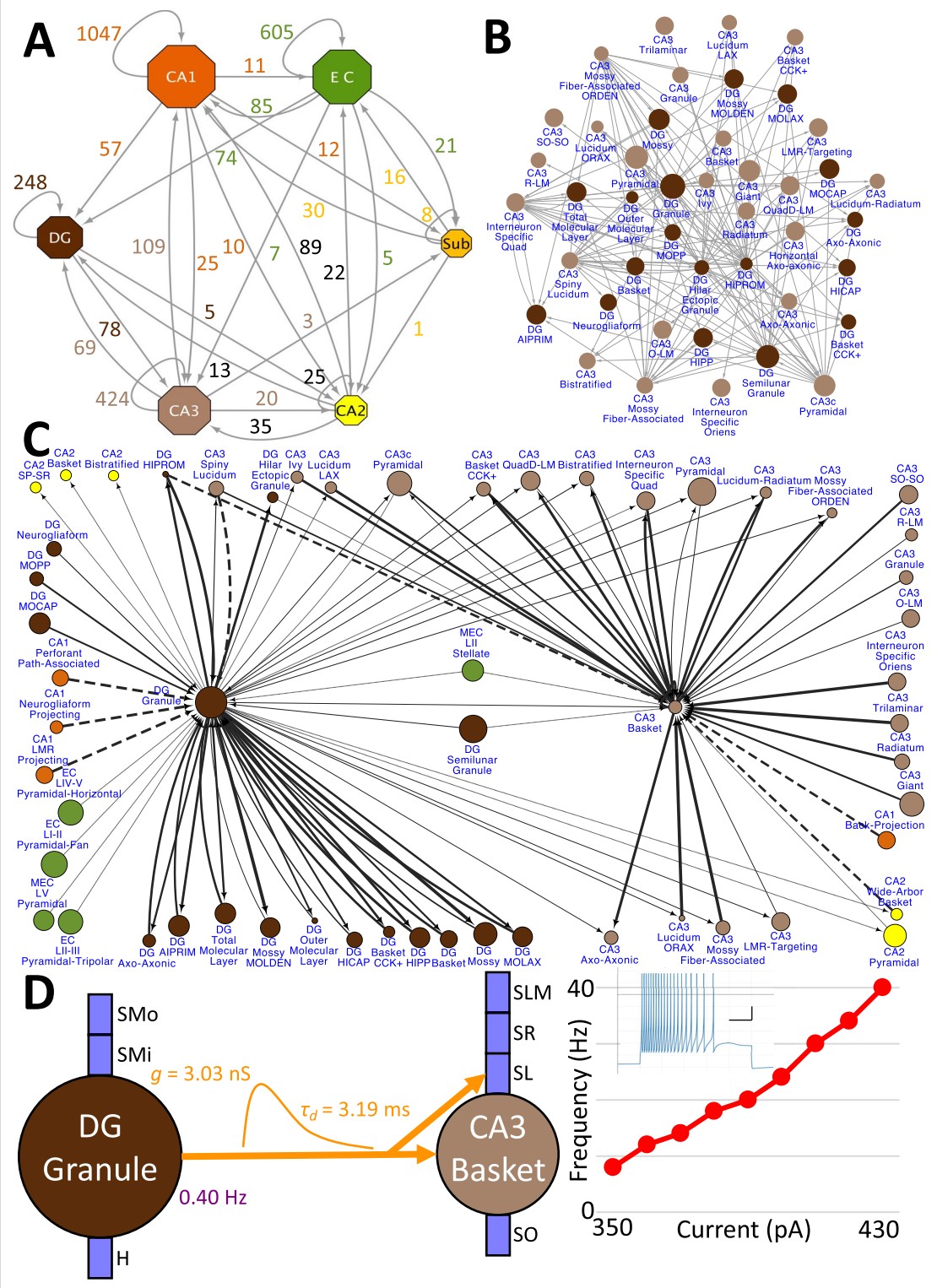

**Figure 8.** Hierarchy of neuronal connectivity in Hippocampome.org. (**A**) Subregional connectivity, where the number of connections between subregions is shown, and the node size is proportional to the number of neuron types in each subregion. (**B**) The reciprocal connectivity between DG and CA3 neuron types consists of 147 connections. The node size is proportional to the census size for each neuron type. (**C**) The full connectivity involving DG Granule and CA3 Basket neuron types consists of 98 connections. The node size is proportional to the census size for each neuron type, and the thicknesses of the connecting arrows are proportional to the synaptic probability. The dashed lines are connections for which the synaptic probability has been approximated based on the means of known values. (**D**) The electrophysiological connection between a DG Granule cell and a

*Figure 8 continued on next page*

*Figure 8 continued*

CA3 Basket cell. The in vivo firing rate is shown for the presynaptic neuron. The transfer function between the two neuron types is proportional to the synaptic conductance times the single-exponential decay time constant ($g \cdot \tau_d$; rat, male, P56, 32 °C, current clamp). The frequency-current (F–I) curve of the single-compartment Izhikevich model of a CA3 Basket cell was obtained with 10 pA current steps. Inset: Izhikevich model firing pattern of a CA3 Basket cell simulated with 430 pA of current applied for 500 ms (vertical and horizontal scale bars, respectively).

spiking neural network simulations to investigate computational theories of cognitive functions in hippocampal circuits at the level of biologically detailed mechanisms.

## Discussion

Hippocampome.org, through its continuous updates and conspicuous usage, has established itself prominently amongst other readily accessible, evidence-based, expert-curated bioscience public resources of note, such as FlyBase for *Drosophila* molecular biology (**The FlyBase Consortium, 1994**; **dos Santos et al., 2015**), WormBase for nematode genomics (**Stein et al., 2001**), the Blue Brain Project for somatosensory cortex (**Markram, 2006**), SynGO for synaptic functions (**Koopmans et al., 2019**), and RegenBase for spinal cord injury biology (**Callahan et al., 2016**). Hippocampome.org has evolved from being a storehouse of information in v1.0–1.12, along the lines of FlyBase, WormBase, SynGO, and RegenBase, to a platform in v2.0 for launching detailed simulations of the hippocampal formation, in the vein of the Blue Brain Project. However, Hippocampome.org distinguishes itself in its reliance wholly on already published data and the more tailored focus on a single portion of the brain.

The foundation for Hippocampome.org has always been the data that are published in the literature. Although a certain level of interpretation is always necessary to make the data machine readable and suitable for database incorporation, data inclusion does not depend on how the data are modeled. Nevertheless, some of the simulation-ready parameters now also included in Hippocampome.org are indeed the result of modeling, such as the neuronal input/output functions (Izhikevich model; **Izhikevich, 2003**) and the unitary synaptic values (Tsodyks-Pawelzik-Markram model; **Tsodyks et al., 1998**). Other simulation-ready parameters are the result of specific analysis approaches, including the connection probabilities (axonal-dendritic spatial overlaps) and the neuron type census (numerical optimization of all constraints).

The growth of Hippocampome.org since the initial release of v1.0 (**Wheeler et al., 2015**) has been prodigious. To date, the site has been visited over 136,000 times with over 33,000 unique visits, and the original publication has been cited more than hundred times. Each successive release of Hippocampome.org v1.X has added new dimensions of knowledge and/or functionality and has been building toward assembling all the components necessary to produce real-scale computational models of the rodent hippocampal formation. The culmination of all this work is the release of Hippocampome.org v2.0, which introduces a framework for launching computer simulations directly from the accumulated knowledge. However, achieving simulations does not mark the end point for this project, because Hippocampome.org will continue to aggregate new knowledge as it is published in the peer-reviewed literature. Gradually, the focus of this resource will shift from development to exploitation through the in silico emulation of complex dynamics observed in vivo and in vitro, with the goal of shedding light on the underlying synaptic-level computational mechanisms.

The creation of real-scale spiking neural network models of the hippocampal formation and its subregions can foster biologically realistic, data-driven, mesoscopic simulations of cognitive function and dysfunction (**Sutton and Ascoli, 2021**). For instance, simulations with Hippocampome.org's real-scale model of the dentate gyrus can build on previous network models of epileptogenesis (**Dyhrfjeld-Johnsen et al., 2007**) by providing further clarity to the roles of all documented neuron types and their corresponding potential connections in seizure initiation and propagation. A real-scale model of CA1 can aim to further the insights into the spatiotemporal dynamics of the circuit during theta oscillations (**Bezaire et al., 2016**; **Navas-Olive et al., 2020**; **Romani et al., 2023**). Furthermore, network models involving multiple subregions can open new vistas on unexplored territories, such as the use of real-scale models of the entorhinal cortex and CA2 to simulate the neuron- and connection-type specific mechanisms of social memory (**Lopez-Rojas et al., 2022**). Moreover, open source sharing of the real-scale models replicating those functions (**Gleeson et al., 2017**) will facilitate cross-talk within

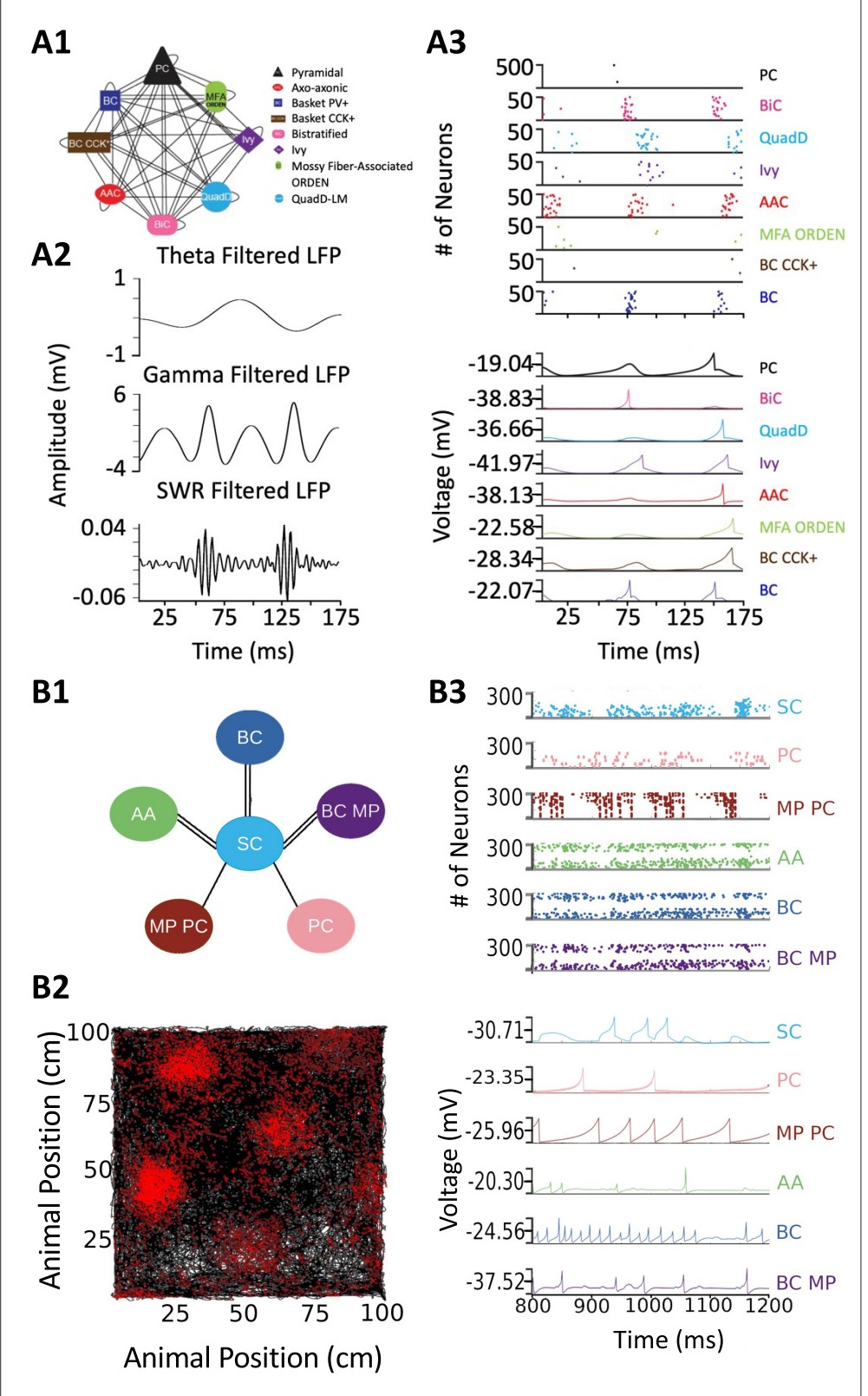

**Figure 9.** Spiking neural network simulations. (**A**) Full-scale CA3 model. (**A1**) Neuron type connectivity schematic. (**A2**) Theta (4–12 Hz; top), Gamma (25–100 Hz; middle), and Sharp-Wave Ripple (150–200 Hz; bottom) filtered local field potentials from 175ms of the simulation. (**A3**) Raster plot of 500 Pyramidal cells and 50 interneurons of each type (top), and representative voltage traces for each neuron type (bottom) during the same 175ms of the

*Figure 9 continued on next page*

*Figure 9 continued*

simulation in (**A2**). (**B**) A mock up of a spatial representation through grid cell firing. (**B1**) Neuron type connectivity schematic. (**B2**) Simulated animal trajectory (black) with red dots indicating the firing of a neuron in those locations. (**B3**) Raster plot of 300 neurons from each type (top), and representative voltage traces for each neuron type. Abbreviations by panel: (**A**) PC: Pyramidal cell; BiC: Bistratified cell; QuadD: QuadD-LM cell; AAC: Axo-axonic cell; MFA ORDEN: Mossy Fiber-Associated ORDEN cell; BC CCK+: cholecystokinin-positive Basket cell. (**B**) SC: Medial Entorhinal Cortex Layer II Stellate cell; PC: Entorhinal Cortex Layer III Pyramidal cell; MP PC: Entorhinal Cortex Layer I-II Multipolar Pyramidal cell; AA: Entorhinal Cortex Layer II Axo-axonic; BC: Medial Entorhinal Cortex Layer II Basket; BC MP: Entorhinal Cortex Layer II Basket-Multipolar Interneuron.

the systems neuroscience community to better understand the role of distinct neuron types in hippocampal function.

A notable aspect of Hippocampome.org is that all freely downloadable model parameters are directly linked to the specific peer-reviewed empirical evidence from which they were derived. Thus, if users disagree with a specific interpretation, or are not fully convinced by an individual experimental measurement, they maintain control in selecting the information sources. Conversely, researchers can choose to reuse the collated experimental data to constrain different computational models they may prefer, such as adopting the Hodgkin-Huxley formalism instead of Izhikevich dynamics. At the same time, Hippocampome.org is not only a collection of model parameters and corresponding empirical evidence, but it also provides an opportunity to unearth knowledge gaps, as facilitated by an intuitive search functionality (hippocampome.org/find-neuron). Missing data can serve to guide the design of targeted 'low hanging fruit' experiments or to generate new hypotheses.

Another important element of Hippocampome.org is the careful annotation of the experimental metadata for each piece of evidence, including the species (rat or mouse), sex (male or female), age (young or adult) as well as any and all reported details that could affect the recorded neuronal property. Examples of these confounding factors abound especially for in vitro electrophysiological data, such as the exact chemical composition of the solution in the electrode and in the bath, slice thickness and orientation, clamping configuration, recording temperature, and animal weight. Because these covariates, when reported by the original investigators, are also stored in the database, it is possible to account for them in subsequent analyses and simulations. Hippocampome.org therefore constitutes a considerably rich one-stop resource to compare and 'translate' key parameters, such as the amplitude and duration of a synaptic signal between two specifically identified neuron types, for instance, from 14-day-old male rat at 22 °C in voltage clamp to a 56-day-old female mouse at 32 °C in current clamp. When fed into spiking neural network simulations, these differential parameter values can foster intuition while attempting to reconcile neuroscience theories and observations.

Hippocampome.org is yet poised for the onset of an information deluge from current and future big science projects, which will need to be integrated into a complete cohesive picture (*de la Prida and Ascoli, 2021*). Although morphological identification will continue to play a fundamental role in defining neuron types and circuit connectivity, the manner in which knowledge is cross-referenced in this resource will allow its effective linkage to rapidly accumulating molecular and imaging data. The ongoing spatial transcriptomics revolution is already transforming the frontiers of cellular neuroscience, often using the hippocampus as its favorite sandbox (*Lein et al., 2017*; *Yao et al., 2021*; *Zeisel et al., 2015*). Single-cell transcriptomics via scRNAseq can bolster the current morphological information by offering distinct transcription factor codes for existing neuron types and assist in defining new ones (*Cembrowski and Spruston, 2019*; *Winnubst et al., 2020*; *Yuste et al., 2020*). From the functional side, optical imaging via genetically encoded voltage indicators (*Knöpfel and Song, 2019*) will provide in vivo voltage traces for defined neuron types that can greatly enhance the repertoire of firing pattern phenotypes to utilize in simulations (*Adam et al., 2019*). Data-driven computational models can provide a useful conceptual bridge between molecular sequencing and activity imaging by investigating the effects of specific subcellular distributions of voltage- and ligand-gated conductances on neuronal excitability (*Migliore et al., 2018*). With the converging maturation of these young techniques and the advent of others yet on the horizon, Hippocampome.org will be able to integrate multidimensional knowledge on the solid foundation of neuronal classification.

## Materials and methods

### Hippocampome.org v2.0 vs. the legacy status of v1.12

With the release of v2.0 of Hippocampome.org upon publication of this article, v1.12 of the website will no longer be updated and will transition to legacy status (hippocampome.org/legacy_v1). In this way, users may avail themselves of the full benefits of the new content and functionality of v2.0, while maintaining access to reference content as published through v1.12. In the near term, neuron types new to v2.0 are tagged with an asterisk on the web site to differentiate them from v1.X types.

### Linking neuron types to NeuroMorpho.Org morphological reconstructions

Hippocampome.org neuron types are regularly linked to appropriately identified digital reconstructions of neuronal morphology from NeuroMorpho.Org (*Ascoli et al., 2007*). Identification of suitable reconstructions with individual neuron types depends on the correspondence of dendritic and axonal locations across hippocampal subregions and layers, as they appear in the reference publication. Alternatively, direct cell typing by the authors in the reference publication text is accepted as evidence for canonical (principal cell) types, such as CA1 Pyramidal cells or Dentate Gyrus Granule cells. Reconstructions are not linked to a neuron type if the experimental conditions are inconsistent with the inclusion criteria of Hippocampome.org, as in the case of cell cultures or embryonic development. Lack of either axonal or dendritic tracing also disqualifies reconstructions of non-canonical neurons from being linked.

### Connections from DG Granule cells to CA3

To compute estimates of connection probabilities and numbers of contacts per connected pair for the rat mossy fiber-CA3 circuit, we used previously calculated average convex hull volume (*Tecuatl et al., 2021b*) and several measurements from a seminal anatomical study (*Acsády et al., 1998*): DG Granule cell axonal length within CA3 (3,236 µm), inter-bouton distances for mossy boutons on Pyramidal cell targets in CA3c (162 µm) and in the rest of CA3 (284 µm), and inter-bouton distances for en-passant and filipodia boutons onto CA3 interneurons (67.4 µm, considering that 48 interneurons can be contacted by a single GC). Given that the mossy fibers innervate mainly CA3 SL, and due to the lack of information regarding the exact proportion of axons innervating CA3 SP, these calculations assume that GCs only innervate SL. The probabilities of connection and numbers of contacts per connected pair (*Table 2*) are estimated as previously described (*Tecuatl et al., 2021a*) utilizing the CA3 dendritic lengths reported in Hippocampome.org.

### Connections from CA3 and CA3c Pyramidal cells to CA1

To compute estimates of connection probabilities and numbers of contacts per connected pair for the rat Schaffer collaterals-CA1 circuit, we utilized previously reported values for the distinct axonal innervation patterns (*Ropireddy et al., 2011*; *Sik et al., 1993*; *Wittner et al., 2007*) in CA1 *stratum radiatum* (SR) and *stratum oriens* (SO) from CA3 Pyramidal cells (27.5% of total axonal length: 64% to SR, 15% to *stratum pyramidale* (SP), 21% to SO) and CA3c Pyramidal cells (64.1% of total axonal length: 94% to SR, 3% to SP, 3% to SO). In addition, we used the average inter-bouton distance reported for the Schaffer collaterals (*Li et al., 1994*) in SR (4.47 µm) and SO (5.8 µm). Total axonal length was measured with L-Measure (*Scorcioni et al., 2008*) from three NeuroMorpho.Org reconstructions for CA3c (NMO_00187, NMO_00191) and CA3b (NMO_00931). We extracted parcel-specific convex hull volumes from Janelia MouseLight (*Winnubst et al., 2019*) Pyramidal cell reconstructions (AA0304, AA0307, AA0420, AA0960, AA0997, AA0999, AA1548) mapped to the 2022 version of the Allen Institute Common Coordinate Framework (CCF). The probabilities of connection and number of contacts per connected pair were estimated as previously described (*Tecuatl et al., 2021a*) using CA1 dendritic lengths from Hippocampome.org. We used separate values for inter-bouton distances in CA1 SR for CA3c Pyramidal cells (5.5 µm: *Wittner et al., 2007*) and CA3 Pyramidal cells (3.7 µm: *Shepherd et al., 2019*; 4.4 µm: *Li et al., 1994*; 4.29 µm: *Sik et al., 1993*; averaged as 4.1 µm).

## Constructing Hippocampome.org spiking neural simulations

Hippocampome.org utilizes CARLsim (*Nageswaran et al., 2009*) as its default simulation environment (https://sites.socsci.uci.edu/~jkrichma/CARLsim/). CARLsim is a graphics processing unit (GPU)-accelerated library of functions for simulating spiking neural networks based on Izhikevich neuron models (*Izhikevich, 2003*). We selected CARLsim due to this ability to run on collections of GPUs, as the power of a GPU supercomputer is needed to simulate the millions of neurons that comprise a full-scale spiking neural network simulation of the complete hippocampal formation. The current version is CARLsim 6 (*Niedermeier et al., 2022*), and the most up-to-date Hippocampome.org-optimized code base, including features not yet released in the main CARLsim version, can be found at hippocampome.org/CARLsim (*Kopsick et al., 2023*).

## Web portal, database, and source code

Hippocampome.org runs on current versions of Chrome, Safari, and Edge web browsers, and it is deployed on a CentOS server running Apache. The website runs off of PHP from a MySQL database. The code for Hippocampome.org v2.0 is available open source at GitHub (copy archived at *Wheeler et al., 2023*). This includes all code for displaying the pages of the website, all scripts for importing spreadsheets into the database, code for using evolutionary algorithms to optimize Izhikevich model parameters, code for the graph theory analysis of the potential connectome, code for the implementation of the firing pattern classification algorithm, and code for analyzing network simulations in CARLsim.

## Acknowledgements

We thank David J Hamilton, Charise M White, Christopher L Rees, Maurizio Bergamino, Keivan Moradi, Siva Venkadesh, Alberto Sanchez-Aguilera, Teresa Jurado-Parras, Manuel Valero, Miriam S Nokia, Elena Cid, Ivan Fernandez-Lamo, Daniel García-Rincón, Liset M de la Prida, Sarojini M Attili, Iqbal Addou, and the many student interns (hippocampome.org/thx) for invaluable help advancing from v1.0 to v2.0. This work was supported in part by grants R01NS39600, RF1MH128693, and U01MH114829 from the National Institutes of Health (NIH). The funding sources were not involved in study design, data collection and interpretation, or the decision to submit the work for publication.

## Additional information

### Funding

| Funder | Grant reference number | Author |
| --- | --- | --- |
| National Institutes of Health | R01NS39600 | Giorgio A Ascoli |
| National Institutes of Health | RF1MH128693 | Giorgio A Ascoli |
| National Institutes of Health | U01MH114829 | Giorgio A Ascoli |

The funders had no role in study design, data collection and interpretation, or the decision to submit the work for publication.

### Author contributions

Diek W Wheeler, Conceptualization, Data curation, Software, Supervision, Investigation, Visualization, Methodology, Writing – original draft, Project administration; Jeffrey D Kopsick, Data curation, Software, Formal analysis, Investigation, Visualization, Methodology, Writing – review and editing; Nate Sutton, Data curation, Software, Investigation, Visualization, Methodology, Writing – review and editing; Carolina Tecuatl, Data curation, Formal analysis, Supervision, Validation, Investigation, Visualization, Methodology, Writing – review and editing; Alexander O Komendantov, Formal analysis, Visualization, Writing – review and editing; Kasturi Nadella, Software, Writing – review

and editing; Giorgio A Ascoli, Conceptualization, Resources, Data curation, Supervision, Funding acquisition, Investigation, Visualization, Methodology, Project administration, Writing – review and editing

## Author ORCIDs
Diek W Wheeler  https://orcid.org/0000-0001-8635-0033
Jeffrey D Kopsick  https://orcid.org/0000-0002-8175-0246
Nate Sutton  http://orcid.org/0000-0002-4424-3886
Carolina Tecuatl  http://orcid.org/0000-0001-6398-3444
Alexander O Komendantov  http://orcid.org/0009-0005-2360-4805
Kasturi Nadella  http://orcid.org/0009-0008-5905-5328
Giorgio A Ascoli  https://orcid.org/0000-0002-0964-676X

Reviewer #2 (Public Review): https://doi.org/10.7554/eLife.90597.3.sa1
Reviewer #3 (Public Review): https://doi.org/10.7554/eLife.90597.3.sa2
Author Response https://doi.org/10.7554/eLife.90597.3.sa3

## Additional files

### Supplementary files
• MDAR checklist

### Data availability
The current manuscript is a description of an updated resource, so no data have been generated for this manuscript. The source code for our resource is available from GitHub (copy archived at *Wheeler et al., 2023*).

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
