## [Editor Report · eLife assessment]

The authors have greatly expanded their **important** hippocampome.org resource about rodent hippocampal cell types, their physiological properties, and their interactions. With version 2.0, they make a significant advance in providing a user-friendly means to make computer models of hippocampal circuits. The work is **convincing**, and there are only minor reservations that the figures may be too complex.

---

## [Referee Report · Reviewer #2 (Public Review)]

The authors have greatly expanded their helpful hippocampome.org resource for the community regarding hippocampal cell types and their interactions from many perspectives. The many updates from v1.0 to v1.12 are nicely summarized in Table 1.

With v2.0, they now achieve the original vision of their project - to enable data-driven spiking neural network simulations of rodent hippocampal circuits. This work thus moves hippocampome.org from not only being a useful resource but also being able to launch simulations in which the models have direct links to the experimental literature. This will not only be of interest to the vast hippocampal community, but also to the diverse computational neuroscience community as theoretical models can potentially be "experimentally tested" with v2.0 to allow theoretical insights to be more biologically applicable.

---

## [Referee Report · Reviewer #3 (Public Review)]

Summary:

The authors aim to provide a multidisciplinary resource on the structural and physiological organization of the hippocampal system and make the available experimental data available for further theoretical work, providing tools to do so in a very flexible and user-friendly way. Since this is a new version of an already existing data-resource, the authors certainly reach their aim and fulfil expectations that the reader might have. The content of the database is as good as the original data, collected from the published knowledge-database, sometimes with help of the original authors, and the overall quality depends further on how the data are curated by the team of authors and many others who helped them. That process is briefly described and more details are available in descriptions of previous versions and on the website. The data extraction, examples of how data can be used and the part on attempts to model the hippocampus are exiting and open doors to new and exciting research opportunities.

Strengths:

Excellent description with many outlined opportunities. Nicely illustrated and inviting to explore the online database. The database itself is easy to navigate and to access relevant information, allowing to do further research on the available data.

Weaknesses:

The figures are complex, containing a heavy information load. One needs some general knowlegde of the system in order to grasp the enormous potential of what is provided.

---

## [Author Response]

The following is the authors’ response to the original reviews.

**Reviewer #2 (Recommendations For The Authors):**
While the details are mostly well-explained, I think that the authors could better bring forth the goals and potential usages of hippocampome.org overall.I think that this is a great and helpful tool that can leverage various and detailed cellular experimental studies that are out there in the literature to garner potential insights, direct future experimental studies, observe/classify experimental 'differences' (e.g., the deep and superficial pyramidal studies they mention) and so on. Say that one gets some mechanistic insight from more abstract theoretical models, hippocampome can be used to determine whether the experimental data where available is supportive of the theory. They also describe CA3 model and grid cells. While I am not suggesting that the authors completely re-organize the manuscript, I did feel that the last section 'potential applications...' could have perhaps been brought forth earlier (in a summarized form) for the reader/user to better appreciate hippocampome - indeed it is line 288 that should be near the beginning of the paper I thought.

We thank the Reviewer for the suggestion. We have now included a summary of the simulation readiness of Hippocampome.org in the Introduction.

I thought the 'application' paragraph (starting line 288) needed expansion to appreciate - I did not have a chance to look at the cited papers in that section - but maybe 2 paragraphs, one on CA3 and the other on grid cells, with a few more sentences of goal/context and tool usage details could be provided?

We thank the Reviewer for the suggestion. We have added expanded paragraphs describing the simulation work on CA3 and grid cells.

The authors start their Discussion by mentioning other resources (e.g. blue brain) in comparison. I thought that this was not too helpful without a bit more expansion about these other resources and what in particular is comparable. For example, the blue brain project is different in that it does not mine the literature per se (I think)? But then I am not sure of the extent of the comparison that the authors intend with blue brain and the other mentioned resources.

Thank you for the helpful suggestion. We have now expanded upon the paragraph to draw more explicit parallels and contrasts among the various projects, in particular between the Blue Brain Project and Hippocampome.org.

Minor commentsFig 3D caption missing

Thank you for pointing this out. We have now amended the figure caption.

Fig 5A line 211-12 refers to v2.0 but Fig 5 caption says v1.0?

We apologize for the confusion. We have now added text clarifying the V1.X relevant descriptions around Figure 5.

Fig 6A confusing with thin and thick arrows and direction?

We apologize for the confusion. We have re-colored the thick arrows orange to emphasize the fact that they are feeding directly into the spiking neural simulations.

Line 260 - not sure what this means - how is importance defined?

We apologize for the confusion. We have now added text clarifying that “importance” refers to the role the neuron type plays in the functioning circuitry of the hippocampal formation.

CARLsim vs Brian/NEST in choosing - maybe a sentence or two for rationale

Thank you for the suggestion. We have now added a sentence explaining the selection of CARLsim. CARLsim was selected due to its ability to run on collections of GPUs. CARLsim was the only simulator with this capability at the time the simulation work was being planned, and the power of a GPU supercomputer was needed to simulate the millions of neurons that comprise a full simulation of the complete hippocampal formation.

Fig 9 mv should be mV, and the voltage values specified there refer to which dash?

Thank you for pointing these situations out. We have amended the millivolts label and have made changes to the figure to help clarify which specific tick marks are being labeled.

**Reviewer #3 (Recommendations For The Authors):**
Compliments to the authors on this nicely organized and structured presentation of V 2.0 of hippocampome.org. The paper is well prepared giving a useful short summary of the history of hippocampome for the newcomers and refreshing the memory of users, switching to highlighting the new data additions, why these are relevant and how these complement the existing database, and opening up to new applications. The added potential is well illustrated and in addition, the authors provide numerical information on the usage of this amazing resource. I enjoyed roaming around in the new version, which was made available for reviewers, and although it has been a while since I worked with the system, the new version is easy to work with. I have not had the time to use it extensively so cannot comment in detail but based on the long experience of the authors and their support team, I trust that version 2 will be almost not completely flawless; however that will for sure become clear when it is released.One could always wish for more, disagree, or even criticize choices made to cluster neurons, divide areas, and so forth, though in my view that does not contribute to what the resource has to offer. Having said this, the authors might consider addressing briefly issues about differences in the nomenclature used in original descriptions and how they handled the translation into their nomenclature. To mention one that is constantly being debated: how does one define the border between SMo and SMi.

Thank you for the suggestion. We have added text to the Introduction that addresses the nomenclature issue, as presented in Hamilton et al. (2017), and provide a definition for SMo and SMi.

Another confusing issue is presented by layers in the entorhinal cortex or its subdivisions (how many and how are these defined). So, some remarks for newcomers in the field who might use the database without spending too much energy to read the original data, might be useful.

Thank you for the suggestion to clarify this situation pertaining to the entorhinal cortex. Often, we have assumed the authors’ own definitions of the layers and subdivisions (medial and lateral), when naming neuron types. When our name is a hybrid of two published names that include both medial and lateral neurons, our name is prefixed by a simple EC, rather than by MEC or LEC.

As noted, the authors present version 2 nicely and comprehensibly and I have only a few additional comments, meant to further improve the already high quality of the paper.1. The figures, nice as they are, are incredibly information-dense, so they require serious study to get the details; the legends do help, but the many abbreviations coming from totally different fields make it challenging to keep track of them while reading. This is a pity since there is a lot of new information in this version of the dataset, compared to previous versions and the authors overall succeed in emphasizing what is new and why this might be of use/importance.

So a few suggestions: (i) add relevant/most important abbreviations to the legends of the individual figures; (ii) introduce all abbreviations upon first use and do not simply refer to the table in the methods. Interestingly, even the authors lose track in the introduction where they use BICCN in line 43 and refer to the abbreviation list, though the full name is given two lines below.

We apologize for the confusion. We have amended the main text to clarify abbreviations. We have added the abbreviation definitions to the captions of the figures, and in some instances, removed the abbreviations from the figures altogether where space allowed.

1. Figure 3 and even more so figure 5 depend strongly on the color differences red/green; please change since generally red/green is no longer used for obvious reasons.

Thank you for pointing this out. We have switched the fonts in Figure 3 to black (excitatory) and gray (inhibitory) to match our previous publication. We have also changed the color schemes in Figure 5 to avoid red and green.

Reviewer #3 commented on the complexity of our figures and how the figures are information dense. To partially address this, we have decided to remove panel A2 of Figure 3. It was originally meant to emphasize where the information came from to add new axonal projections to two v1.0 neuron types; however, it is not necessary to make the point in the illustration. Thus, we have removed the panel and amended the caption for Figure 3A to include the cited reference.